# The Development of a Gleason Score-Related Gene Signature for Predicting the Prognosis of Prostate Cancer

**DOI:** 10.3390/jcm11237164

**Published:** 2022-12-01

**Authors:** Yiliyasi Yimamu, Xu Yang, Junxin Chen, Cheng Luo, Wenyang Xiao, Hongyu Guan, Daohu Wang

**Affiliations:** 1Department of Urology, The First Affiliated Hospital of Sun Yat-sen University, Guangzhou 510060, China; 2Department of Endocrinology and Diabetes Center, The First Affiliated Hospital of Sun Yat-sen University, Guangzhou 510060, China

**Keywords:** prostate cancer, radical prostatectomy, gene signature, risk score, biochemical recurrence, Gleason score

## Abstract

The recurrence of prostate cancer (PCa) is intrinsically linked to increased mortality. The goal of this study was to develop an efficient and reliable prognosis prediction signature for PCa patients. The training cohort was acquired from The Cancer Genome Atlas (TCGA) dataset, while the validation cohort was obtained from the Gene Expression Omnibus (GEO) dataset (GSE70769). To explore the Gleason score (GS)-based prediction signature, we screened the differentially expressed genes (DEGs) between low- and high-GS groups, and then univariate Cox regression survival analysis and multiple Cox analyses were performed sequentially using the training cohort. The testing cohort was used to evaluate and validate the prognostic model’s effectiveness, accuracy, and clinical practicability. In addition, the correlation analyses between the risk score and clinical features, as well as immune infiltration, were performed. We constructed and optimized a valid and credible model for predicting the prognosis of PCa recurrence using four GS-associated genes (SFRP4, FEV, COL1A1, SULF1). Furthermore, ROC and Kaplan–Meier analysis revealed a higher predictive efficiency for biochemical recurrence (BCR). The results showed that the risk model was an independent prognostic factor. Moreover, the risk score was associated with clinical features and immune infiltration. Finally, the risk model was validated in a testing cohort. Our data support that the GS-based four-gene signature acts as a novel signature for predicting BCR in PCa patients.

## 1. Introduction

Prostate cancer (PCa) is believed to become the second most common type of malignancy in men [1,2]. Measuring prostate-specific antigen (PSA) in the blood is a very sensitive way to detect an increased risk of lethal disease at a very early stage. The proportion of PCa is increasing rapidly each year in China. According to cancer statistics for 2020, 115,426 individuals were diagnosed with PCa and 51,095 succumbed to the malady in China [3], a level that has nearly doubled since 2015 [4]. For patients with localized curative PCa, therapeutic options such as radical prostatectomy, external beam radiotherapy, and low-dose brachytherapy are primarily recommended. The standard treatment for metastatic disease is androgen deprivation therapy [5]. The condition of an increased serum PSA level following radical prostatectomy (RP) or radiation treatment for localized PCa is known as biochemical recurrence (BCR) or biochemical relapse [6]. BCR signifies a higher risk of metastases and PCa that is resistant to castration [7]. BCR is characterized by two or even more consecutively elevated serum PSA levels of more than 0.2 ng/mL for patients who underwent radical prostatectomy [8]. Within 5 years of receiving initial therapy, approximately 20 to 30 percent of patients with clinically localized cancer will experience a clinical recurrence [9,10]. Clinical recurrence and metastasis are more likely to occur in BCR patients, especially early BCR patients, while the median survival for patients presenting with metastatic PCa is only 30 months [11]. Therefore, it is essential to discover more accurate biomarkers for the better prediction of BCR and monitoring the disease.

Tumor biomarkers used in combination with existing clinical predictors of recurrence such as Gleason score, stage, and preoperative PSA level [12,13] have been shown to provide additional information for accurately differentiating aggressive and indolent PCa patients [14]. With the advent of gene chips and high-throughput sequencing, it has become a fact that gene signatures formed from aberrant transcriptional patterns may predict prognosis for PCa patients. Multiple gene signatures for metastatic PCa prognosis prediction have been proposed in the last few years. In the case of PCa with heterogeneous hypoxia, Yang et al. proposed a 28-gene hypoxia-related signature as a powerful predictor of 5-year BCR-free survival (BFS) in PCa patients after RP or receiving post-prostatectomy radiotherapy [15]. Several studies [16] used similar methods to generate mRNA expression signatures with varying numbers of genes. Meanwhile, some aberrant gene expression information has been verified in subsequent studies, providing critical information for clinical treatment decisions [17,18]. There is thus a need to identify novel signatures that can distinguish patients at high risk for BCR.

Here, we aim to investigate a novel signature based on public data from The Cancer Genome Atlas (TCGA) database. TCGA has generated a huge amount of cancer data using high-throughput sequencing techniques and is supervised by the National Cancer Institute and the National Human Genome Research Institute [19]. The GS is the sum of the grades of the first and second Gleason patterns of a primary cancer sample. It has been clearly shown that GS have been associated with BCR [20] and prostate cancer mortality [21]. Considering GS is one of the best predictors of PCa prognosis, the strategy of finding differentially expressed transcripts in tumors stratified by GS is anticipated to acquire pertinent data on the potential for tumor aggressiveness [22]. In this study, we constructed a GS-based four-gene signature and evaluated its ability to predict recurrence after radical prostatectomy in PCa patient cohorts.

## 2. Materials and Methods

### 2.1. Gene Expression Profile and Clinical Data

Gene expression datasets in FPKM and Counts formats with the clinic information of 487 PCa samples were collected from TCGA PCa dataset (https://portal.gdc.cancer.gov/, accessed on 1 June 2022) using the TCGA bio-links package, which was used for training set. Among these patients, 410 patients with complete BCR information were utilized for BCR-free survival. Another gene expression dataset (GSE70769) was mined from the Gene Expression Omnibus (GEO) database (https://www.ncbi.nlm.nih.gov/geo/, accessed on 1 June 2022) and used as the validation cohort. The overall bioinformatic analysis workflow in this study is presented in Figure 1.

### 2.2. Analysis and Screening of Differentially Expressed Genes (DEGs) between Low- and High-GS Groups

Among the 487 PCa patients, there were 291 cases with GS < 8 (low-GS group) and 196 cases with GS ≥ 8 (high-GS group). The patients’ clinical characteristics of the training dataset are shown in Appendix A. We analyzed the DEGs between low- and high-GS groups with the R package DEseq2, and the cut-off was set to adjusted *p* < 0.05 and |log2(fold change)| ≥ 0.5. The volcano map was developed using ‘ggplot2′ R package. Considering the biological significance and the convenient for further detection, those genes with mean FPKM less than 10 in tumor tissues were excluded from GS-associated DEGs.

### 2.3. Gene Oncology (GO) Analysis

We obtained the GO terms enrichment using ‘clusterProfiler’ R package with function ‘enrichGO’. The top 10 GO terms enriched in biological processes (BP) were visualized by ‘dotplot’ function.

### 2.4. Construction and Validation of the GS-Associated Gene Signature

BCR-free-survival-related genes were screened using Cox regression analysis based on the previously identified DEGs. We performed univariate Cox analysis using ‘Survival’ package in R and selected genes with *p* < 0.05 for next multivariate Cox regression analysis. The risk score was calculated using following formula: β1* Exp1 + β2* Exp2 + …. + βi* Expi. The β values were acquired by multivariate Cox regression analysis, and Exp was the level of indicated genes’ expression. The optimal cut-off values that could classify the patients in each dataset into high-risk and low-risk groups were obtained using ‘survivalROC’ package in R. Kaplan–Meier curves were plotted using ‘survfit’ function in ‘survival’ package, and *p* values were calculated with log rank test. The ‘survivalROC’ package of R software was performed to investigate the time dependence of the risk model.

### 2.5. Independent Prognostic Analysis

In order to assess whether risk score could be an independent risk factor, we performed univariate and multivariate Cox regression model analyses in training and testing cohorts. Variables with *p* < 0.05 in the multivariate Cox regression analysis were identified as independent risk factors.

### 2.6. Immune Microenvironment Analysis

Two deconvolution algorithms, namely CIBERSORT [23] and ESTIMATE [24], were performed to analyze the tumor microenvironment contexture. The relationship between the risk score and immune microenvironment was then assessed.

### 2.7. Box Plotting

Box plots were prepared using ‘ggboxplot’ function in the ‘ggpubr’ package of the R program. ‘Stat_compare_means’ function in the ‘ggpubr’ package of R program was used to determine the significance.

### 2.8. Statistical Analysis

Student *t* test was used to compare the expression of genes and risk score between indicated two groups. The BCR-free survival between indicated two groups was compared by Kaplan–Meier analysis, and *p* values were calculated with log rank test. Univariate Cox analysis and multivariate Cox regression analysis were performed to explore independent predictors of BCR-free survival. The hazard ratio (HR) and 95% confidence interval (CI) were calculated to compare the associated with BCR-free survival. ROC curves were implemented to identify the predictive accuracy of the risk signatures. All analyses were processed using the R software 4.2.1(The R Foundation, Vienna, Austria) and SPSS Version 26.0 (IBM, Armonk, NY, USA). If not specified above, *p* value less than 0.05 was considered statistically significant.

## 3. Results

### 3.1. The Identification and GO Enrichment Analysis of DEGs between PCa Patients with Low and High GS

Initially, we mined the DEGs based on Gleason score in TCGA-PCa datasets with a total of 487 prostate cancer patients. Among these, 196 cases had high GS (≥8), and 291 cases had low GS (<8). Then, we used |log2FC| ≥ 0.5 and *p* value < 0.05 as the screening criteria. A total of 127 upregulated and 133 downregulated genes were identified (Figure 2A). These upregulated and downregulated genes were used for GO biological processes (BP) analysis. The top 10 GO BP terms are shown in Figure 2B.

### 3.2. Identification of BCR Free Survival-Related Genes and Establishing the Prognostic Prediction Signature

Given that a gene must be expressed to some extent to be considered biologically important, together with the convenient for further detection, we selected 116 genes with mean FPKM greater than 10 in tumor tissues for further univariable regression analysis. We found that 32 genes were significantly associated with BCR-free survival (Table 1). Then, multivariable Cox regression analysis was performed, and a prediction model containing four genes was constructed as follows: (0.06447 ∗ expression of SFRP4) + (−0.06196 ∗ expression of FEV) + (0.37184 ∗ expression of COL1A1) + (0.06892 ∗ expression of SULF1). The risk score of each patient in the training set was calculated by the risk formula. The BCR status, risk score, and expression of the four genes in each patient from the training dataset are shown in Figure 3A,B. We also evaluated the BCR-free survival using the Kaplan–Meier method and log-rank test, and the results (Figure 3C) showed that the BCR-free survival time in the high-GS group was significantly shorter than that in the low-GS group (*p* < 0.0001). Furthermore, a time-dependent ROC curve was performed to evaluate the sensitivity and specificity of the four-gene signature for PFS prediction. Notably, the area under the ROC curve (AUC) was 0.66 for 1-year survival, 0.73 for 3-year survival, and 0.74 for 5-year survival (Figure 3D). Univariate Cox regression analysis and multivariate analysis demonstrated that the risk score was an independent predictor of poor clinical outcome in the training cohort (Table 2).

### 3.3. The Expression of Four Genes in Low-GS and High-GS, as Well as BCR and BCR Free Patients

We next assessed the expression of SFRP4, FEV, COL1A1, and SULF1 in low-GS and high-GS patients. The box plot depicted the expression of four genes in high-GS and low-GS groups. As shown in Figure 4A–D, the expression levels of SFRP4, COL1A1, and SULF1 were significantly upregulated in the high-GS group compared with in the low-GS group, while FEV expression was significantly downregulated in the high-GS group. Of note, the same expression pattern was observed in BCR and BCR-free patients. Among these four genes, FEV has a negative coefficient, indicating that it is a protective factor. SFRP4, COL1A1, and SULF1 have positive coefficients, suggesting negative effects on the prognosis. Intriguingly, FEV was highly expressed in BCR-free patients compared with in BCR patients. In contrast, the levels of SFRP4, COL1A1, and SULF1 were increased in BCR patients (Figure 5A–D).

### 3.4. The Association between Risk Score and Clinical Variables

The correlation between the risk score and clinical features was evaluated. The results showed that pathological T stage (Figure 6A), pathological N stage (Figure 6B), clinical T stage (Figure 6C), age (Figure 6D), and GS (Figure 6E) were significantly correlated with the risk score, indicating that the strong association between risk score and clinical pathological features.

### 3.5. The Correlation between Risk Score and Immune Microenvironment

Firstly, the CIBERSORT algorithm was used to analyze the correlation of risk score with immune signature. The results showed that tumors with high-risk scores were significantly correlated with high fractions of B cell naive, macrophages M1, macrophages M2, T cells CD4 memory resting, and T cells regulatory (Tregs) (Figure 7A). In the group with low-risk scores, there are high fractions of mast cell resting, NK cells activated, and plasma cells (Figure 7A). The abundance of each immune cell subtypes in PCa samples for the low- and high-risk groups is shown in Figure 7B. Moreover, ESTIMATE algorithms were applied to calculate the microenvironment score and immune score. The heatmap of the stromal score and immune score between the high- and low-risk groups is shown in Figure 7D. The box plots indicate that the low-risk group had a lower immune score and stromal score than those in the high-risk group (Figure 7C).

### 3.6. Validation of the Four-Gene Signature Using Another Independent Dataset

To investigate the reliability of the GS-associated four-gene signature, another independent dataset was used for further validation. Among the validation set, patients’ clinical characteristics are shown in Appendix A. We analyzed BCR status, risk score, and the four gene expression levels of each case, and the results are shown in Figure 8A,B. Of specific note, a substantially effective performance for BCR-free survival prediction was observed. Kaplan–Meier analysis exhibited that the patients in the high-risk group had obviously shorter BCR-free survival time than those in the low-risk group (Figure 8C). The AUCs for 1-year, 3-year, and 5-year survival are shown in Figure 8D. These results suggest that the four-gene signature is valid and reliable across datasets and platforms. Additionally, univariate Cox regression analysis and multivariate analysis showed that the risk score was an independent factor for predicting poor clinical outcome in the validation cohort (Table 3).

## 4. Discussion

Currently, PCa recurrence is a popular topic across the world. A large proportion of advanced PCa patients will recur [25], leading to ultimate death. Therefore, it is strongly desirable to effectively discover that patients with PCa have a high risk of BCR following RP. The utility of Gleason score in assessing prognosis at diagnosis is limited by the variability in tumor tissue grading [26], and the substantial heterogeneity of prognosis in patients with GS < 8 tumors [27]. Meanwhile, previous studies have revealed that signatures such as cell cycle progression genes [28], SigMuc1NW [29], lncRNAs [30] and microRNAs [31] have great potential as prognostic biomarkers. However, several concerns limit their prognostic and predictive power, such as inadequate samples and the lack of effective validation. Hence, exploring novel prediction models will greatly improve BCR prediction performance.

Our study predicts the recurrence of PCa after RP based on the GS score. We first identified DEGs between a high-GS group and a low-GS group, and then a univariable Cox analysis was carried out for screening BCR-associated genes. Then we used a multiple Cox regression model to establish a four-gene signature with prediction value. Then the risk score of each patient was calculated according to the formula. We found that BCR-free survival time in the high-GS group was significantly shorter than the low-GS group in each cohort. Time-dependent ROC analyses showed that the risk score had predictive power in both the training and validation sets. Among the four genes, the expression level of the gene (FEV) with negative coefficient was increased in BCR-free patients. Moreover, SFRP4, COL1A1, and SULF1 had positive coefficients, and their expression levels were upregulated in patients with BCR.

Previous studies have implicated these four genes in PCa. SFRP4 is one of the members of the secreted curl-related protein family (SFRP1–5) and is an extracellular inhibitor of Wnt signaling, whose role in carcinogenesis has been determined [32]. For PCa patients, the risk of BCR after RP increased with increased protein level of SFRP4 [33]. Research found that SFRP4 is one of the genes associated with PCa invasiveness and recurrence [33,34]. FEV is part of the ETS transcription factor family. Liang et al. found that its expression was significantly negatively associated with the GS [35]. The high expression of COL1A1 in colon cancer is significantly related to serous membrane infiltration, lymphatic metastasis, and hematogenous metastasis [36]. Liu et al. confirmed that COL1A1 is important in predicting PCa prognosis and progression [37]. Sulfatases (SULFs) have been locally acting tumor suppressors in many carcinomas [38]. Sulfatases 1 (SULF) are closely related to 6-Oendosulfatases [39], which are secreted or remain peripherally associated with proteoglycans at the cell membrane [40]. Some studies show that SULF1 suppresses the Wnt3A-driven growth of bone metastatic PCa [41]. Further studies exploring the clinical and biological significances of these genes in PCa will bring fresh insights into the etiology of the disease.

Bone is the most common site for prostate cancer to metastasize, and the incidence of advanced disease is 65–80% [42]. The skeletal microenvironment facilitates metastasis to PCa due to the complicated interactions between the microenvironment and tumor cells. Aside from cell precursors, bone marrow includes a variety of recirculating mature immune cells, such as dendritic cells (DC), macrophages, several T and B lymphocyte subsets, myeloid-derived suppressor cells (MDSCs), and NK cells. Some of these leukocytes are involved in pathogen clearance as well as anti-tumor actions [43,44]. Using CIBERSORT, we found a significant infiltration promotion of several immune cell subsets, particularly CD4+ T cells and macrophages M2, in high-risk groups. Intriguingly, this phenomenon increases the chance of metastasis and recurrence. In addition, we observed that the high-risk group has a significantly higher immune score and stromal score than the low-risk group, which showed that a distinct tumor microenvironment contexture exists between the two subtypes. Given the critical impact of the tumor immune microenvironment on metastasis, prompt and effective interventions to dysregulated immune cell infiltration can avert tumor recurrence.

There are several limitations in this study that need to be acknowledged. First, as this study was conducted using retrospective data that were obtained from public datasets, further prospective results are needed to support each other. Second, our study had a small number of cases with BCR or who died of PCa. Since BCR occurs over a wide time span, from a few months to over 15 years following the initial treatment [45], a long follow-up period is required to ascertain these outcome events. Third, future study is required to clarify the detailed molecular mechanism and function of these four genes in the development and progression of PCa.

## 5. Conclusions

Taken together, we conducted an integrated study to develop a GS-based four-gene signature (SFRP4, FEV, COL1A1, SULF1) for the prediction of the BCR of PCa patients after RP. The results from this study revealed a powerful prognostic indicator independent of and complementary to existing clinical factors for prognostication in PCa, such as the previously established Partin tables [46] and Kattan nomograms [47].

## Figures and Tables

**Figure 1 jcm-11-07164-f001:**
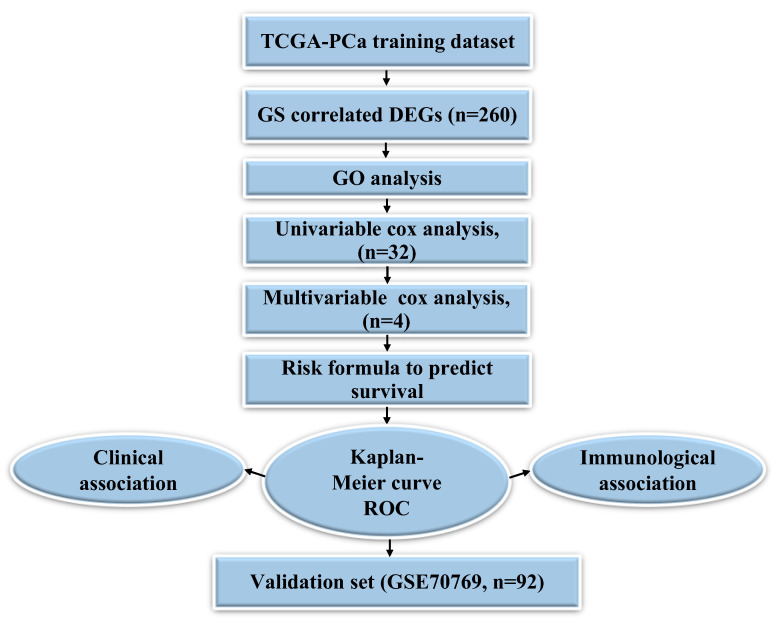
Flow chart of methods for building the Gleason Score (GS) based four-gene signature for prediction of BCR free survival in prostate cancer (PCa) patients; DEGs: differentially expressed genes.

**Figure 2 jcm-11-07164-f002:**
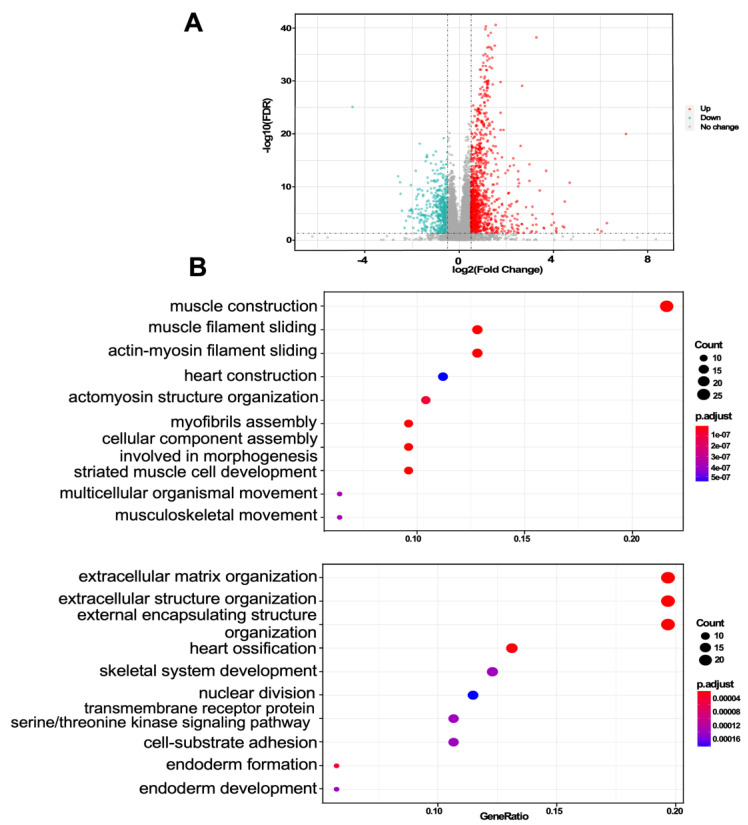
Differentially expressed and functional enrichment analysis. (**A**) Volcano plot of DEGs between PCa patients with low and high GS. Red represents upregulated genes, and blue indicates downregulated genes. (**B**) Gene Ontology biological process analysis of the DEGs.

**Figure 3 jcm-11-07164-f003:**
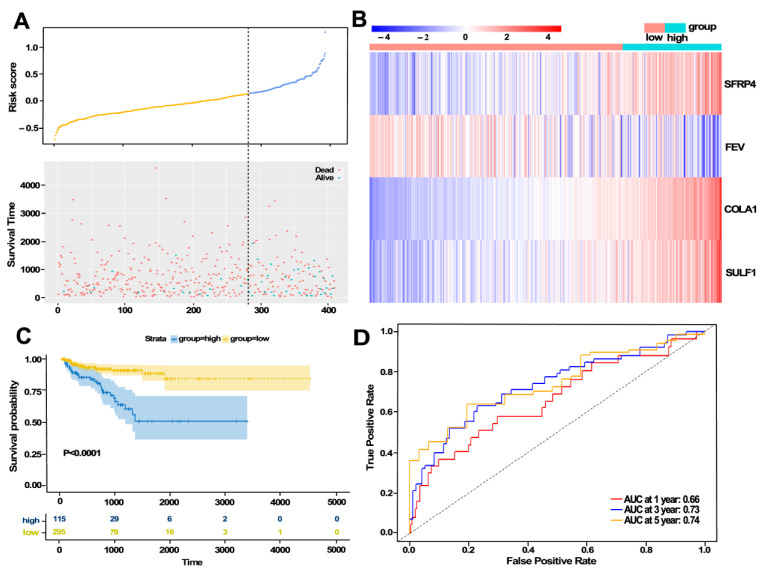
Construction of the four-gene signature in training cohort. (**A**,**B**) Distribution of risk score, BCR status, and gene expression of each case. (**C**) Kaplan–Meier curve analysis. (**D**) ROC curve analysis of the GS based four-gene signatures.

**Figure 4 jcm-11-07164-f004:**
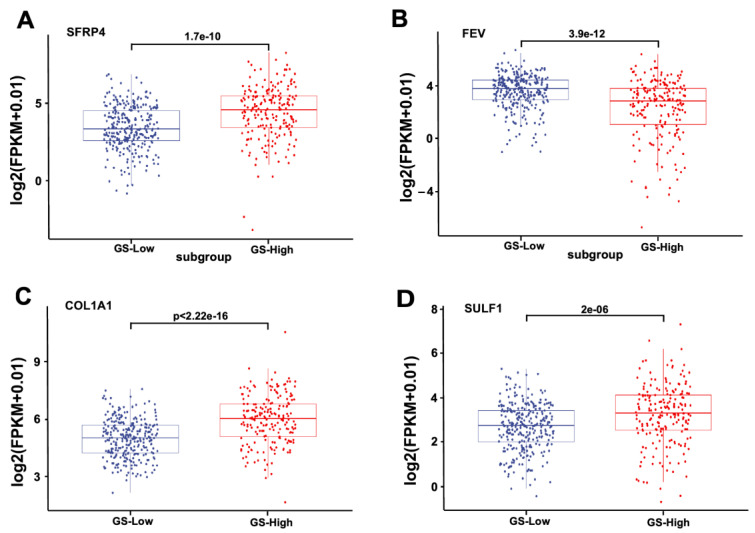
The expression of four genes in high-GS and low-GS groups. The expression of SFRP4 (**A**), FEV (**B**), COL1A1 (**C**), and SULF1 (**D**) in PCa patients with high GS and low GS.

**Figure 5 jcm-11-07164-f005:**
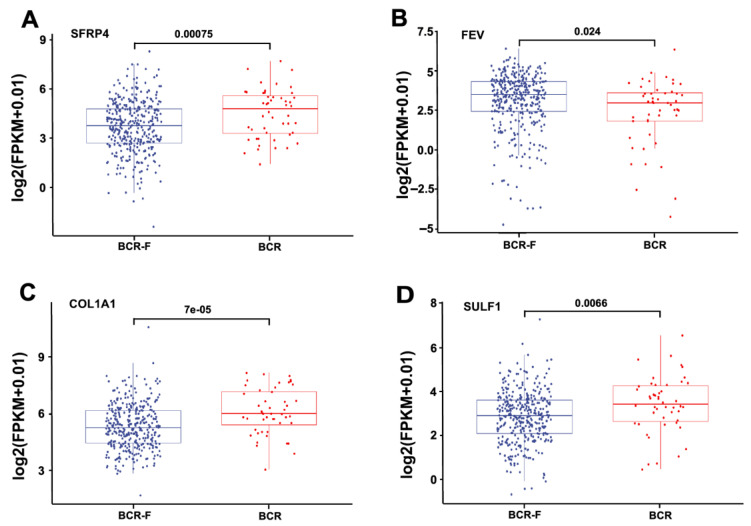
The expression of four genes in BCR and BCR-free groups. SFRP4 (**A**), FEV (**B**), COL1A1 (**C**), and SULF1 (**D**) expression levels in BCR and BCR-free PCa patients.

**Figure 6 jcm-11-07164-f006:**
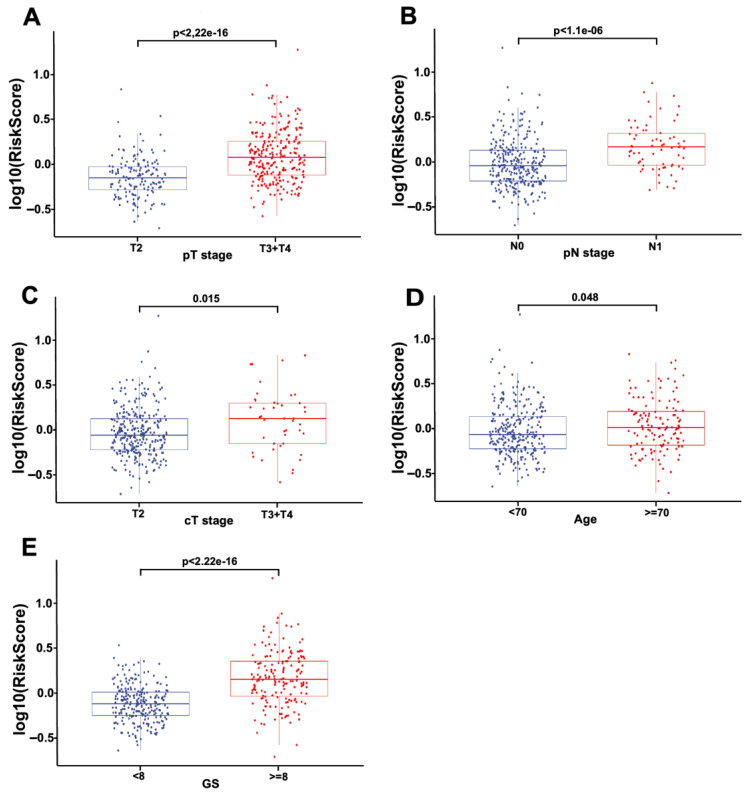
The association between the GS-based four-gene signature and clinical characteristics of PCa patients. The distribution of risk scores was associated with pathologic T (**A**), clinical T (**B**), pathologic N (**C**), Gleason score (**D**), and age (**E**).

**Figure 7 jcm-11-07164-f007:**
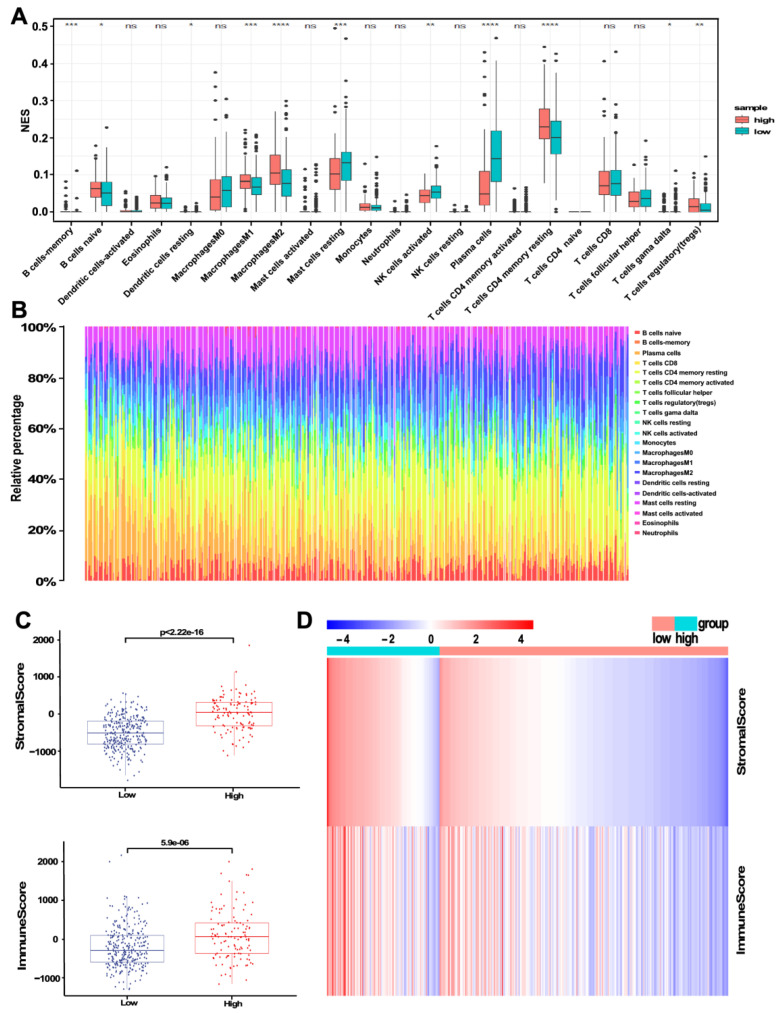
Immune infiltrating cell profile in PCa and correlation analysis. (**A**) The subgroup analysis of 22 immune infiltrating cells between the high-GS and low-GS groups. Red represents high-GS group while green represents low-GS group. (* *p* < 0.05; ** *p* < 0.01; *** *p* < 0.001; **** *p* < 0.0001) (**B**) Bar plot of the proportion of immune infiltrating cells in the high- and low-risk PCa groups. (**C**) Box plots of immune score and stromal scores between the high- and low-risk groups. (**D**) Heat map of immune score and stromal scores between the high- and low-risk groups.

**Figure 8 jcm-11-07164-f008:**
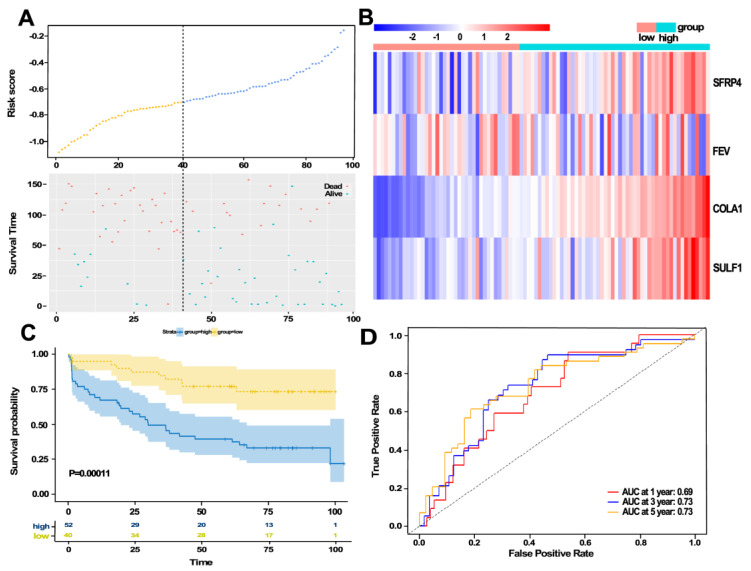
Evaluating the prognostic power of the GS-based four-gene signature in the testing dataset (GSE70769). (**A**,**B**) Distribution of risk score, BCR status and gene expression of every patient in testing cohort. (**C**) ROC curve analysis of GS-based four-gene signature in testing cohort. (**D**) Kaplan–Meier curve analysis in testing cohort.

**Table 1 jcm-11-07164-t001:** Thirty-two genes significantly associated with the BCR free survival of patients in the training set (*n* = 487).

Gene	Hazard. Ratio (CI95)	*p* Value
COL1A1	1.67 (1.37–2.03)	0
BGN	1.9 (1.46–2.46)	1.00 × 10^−6^
SULF1	1.66 (1.34–2.06)	5.00 × 10^−6^
COL3A1	1.65 (1.33–2.04)	6.00 × 10^−6^
COL1A2	1.71 (1.35–2.16)	7.00 × 10^−6^
COMP	1.37 (1.18–1.59)	2.90 × 10^−5^
POSTN	1.55 (1.26–1.91)	3.20 × 10^−5^
IGFBP3	1.71 (1.32–2.23)	6.00 × 10^−5^
ELN	1.7 (1.31–2.22)	8.00 × 10^−5^
SFRP4	1.42 (1.19–1.71)	0.000145
SFRP2	1.51 (1.22–1.87)	0.000151
CDC42EP5	0.67 (0.54–0.83)	0.000221
FN1	1.32 (1.13–1.53)	0.000379
CXCL14	1.33 (1.13–1.56)	0.000473
CAMK2N1	1.37 (1.14–1.63)	0.000532
TK1	1.4 (1.14–1.71)	0.001395
CKS2	1.42 (1.11–1.82)	0.004743
CKM	0.81 (0.7–0.94)	0.005725
RHOU	1.33 (1.08–1.63)	0.006285
MSMB	0.84 (0.74–0.95)	0.006588
LBH	1.28 (1.07–1.54)	0.007336
ALDH1A1	1.2 (1.05–1.38)	0.008648
FEV	0.84 (0.73–0.96)	0.010531
CRIP2	1.36 (1.07–1.72)	0.01076
CHRNA2	0.84 (0.74–0.96)	0.013084
PEBP4	0.85 (0.75–0.98)	0.020598
VSIG2	0.87 (0.77–0.98)	0.022015
KRT14	0.86 (0.75–0.98)	0.029254
AZGP1	0.86 (0.75–0.99)	0.033307
CPE	0.75 (0.57–0.99)	0.040177
LCN2	0.89 (0.79–1)	0.043003
ACP3	0.84 (0.71–1)	0.049737

**Table 2 jcm-11-07164-t002:** Univariate and multivariate analyses of the four-gene-based risk score and clinicopathological characteristics in TCGA-PCa training set.

	Univariable		Multivariable	
Variables	HR (95% CI)	*p*	HR (95% CI)	*p*
Age	1.303 (0.404, 4.203)	0.658		
Gleason Score	0.032 (0.156, 0.584)	<0.01		
pT	0.133 (0.041, 0.429)	0.001	0.218 (0.064, 0.747)	0.015
pN	0.199 (1.078, 3.705)	0.028		
Risk score	0.270 (0.150, 0.487)	<0.01	0.417 (0.222, 0.782)	0.006

**Table 3 jcm-11-07164-t003:** Univariate and multivariate analyses of the four-gene-based risk score and clinicopathological characteristics in GEO70769 validation set.

	Univariable		Multivariable	
Variables	HR (95% CI)	*p*	HR (95% CI)	*p*
Gleason Score	3.716 (1.829, 7.552)	<0.01	2.773 (1.281, 6.005)	0.01
pT	3.915 (2.026, 7.562)	<0.01	2.323 (1.115, 4.838)	0.024
Surgical margins	2.186 (1.185, 4.034)	0.012		
Prostate-specific antigen	1.93 (1.04, 3.582)	0.037	2.254 (1.157, 4.392)	0.017
Risk score	0.331 (0.163, 0.675)	0.002	0.441 (0.206, 0.944)	0.035

## Data Availability

Not applicable.

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
