# Peer review of "The Development of a Gleason Score-Related Gene Signature for Predicting the Prognosis of Prostate Cancer"

_jcm, 2022, doi:10.3390/jcm11237164_

Round 1
Reviewer 1 Report
The authors were able to identify a Gleason score associated 4 gene signature from a database of 260 genes. The signature appears predictive of prognosis in prostate cancer patients and, according to the results presented, represents an independent prognostic factor for predicting biochemical recurrence. Additionally, an association between the gene signature defined risk groups (low and high risk) and the tumor associated immune microenvironment could be shown.
The authors conclude that the gene signature-based prediction model they elaborated may contribute to more accurately predict prognosis and thus help to diagnose patients with biochemical recurrence and make appropriate treatment decisions.
The authors present a very interesting and carefully evaluated study.
Major objectives:
Gleason score 4+3 / ISUP grade group 3 cancers should not be counted as low GS group cancers because they are associated with a significantly worse prognosis than GS 3+4 /ISUP grade group 2 cancers (f.e. Wright JL et al. J Urol 2009, 182:2702-2707). Possibly this would result in an even sharper prediction for the identified gene panel. Perhaps the authors can do some more calculation in this regard or at least address this point.
The authors should critically mention in their conclusion about the possible predictive value of the identified gene signature that the comparison with the established nomograms (for example Partin, Kattan) in this respect still has to be done.
Author Response
- Gleason score 4+3 / ISUP grade group 3 cancers should not be counted as low GS group cancers because they are associated with a significantly worse prognosis than GS 3+4 /ISUP grade group 2 cancers (f.e. Wright JL et al. J Urol 2009, 182:2702-2707). Possibly this would result in an even sharper prediction for the identified gene panel. Perhaps the authors can do some more calculation in this regard or at least address this point.
Response: I appreciate the reviewers' perspectives. We fully agree with you. According to the current prostate cancer guidelines, GS-ISUP grade 3 tumors have a much worse prognosis than GS-ISUP grade 2 malignancies. However, the prognosis for patients with GS-ISUP grade 4 cancers and ISUP grade 5 cancers malignancies is still poorer than for patients with GS-ISUP grade 3 cancers, ISUP grade 2 cancers, and ISUP grade 1 cancers. Meanwhile, numerous studies employ the same grouping method to achieve research results[1-3].Of note, basing on this grouping method, we develop a GS-based four-gene signature (SFRP4, FEV, COL1A1, SULF1) for the prediction of the BCR of PCa patients after RP. The results from this study provide a novel prediction model that, may contribute to more accurately assessing the prognosis and thus help to diagnose patients with BCR and make appropriate treatment decisions.
- The authors should critically mention in their conclusion about the possible predictive value of the identified gene signature that the comparison with the established nomograms (for example Partin, Kattan) in this respect still has to be done.
Response: We thank the reviewer for raising a good point. We have mentioned the predictive value of the identified gene signature in the conclusion part: “Taken together, we conducted an integrated study to develop a GS-based four-gene signature (SFRP4, FEV, COL1A1, SULF1) for the prediction of the BCR of PCa patients after RP. The results from this study revealed a powerful prognostic indicator independent of and complementary to existing clinical factors for prognostication in PCa, such as the previously established Partin tables[4] and Kattan nomograms[5].”
- Kurokawa, R., et al., Osteolytic or mixed bone metastasis is not uncommon in patients with high-risk prostate cancer. Eur J Radiol, 2022. 157: p. 110595.
- Wegener, D., et al., Postoperative Radiotherapy of Prostate Cancer: Adjuvant versus Early Salvage. Biomedicines, 2022. 10(9).
- Nishimoto, M., et al., Prognostic factors in Japanese men with high-Gleason metastatic castration-resistant prostate cancer. Transl Cancer Res, 2022. 11(8): p. 2681-2687.
- Eifler, J.B., et al., An updated prostate cancer staging nomogram (Partin tables) based on cases from 2006 to 2011. BJU Int, 2013. 111(1): p. 22-9.
- Stephenson, A.J. and M.W. Kattan, Nomograms for prostate cancer. BJU Int, 2006. 98(1): p. 39-46.
Reviewer 2 Report
Dear Authors,
I read with interest your article on the search for genes for predicting biochemical recurrence in patients with prostate adenocarcinoma.
The article is interesting but unfortunately has some major limitations:
PRAD is not a correct acronym, please modify in PCa
Line 41: “BCR is characterized by two or even more consecutively elevated serum PSA levels of 41 more than 0.2 ng/ml” it is true just for patients underwent radical prostatectomy, and not radiotherapy
Why is so important the BCR during the follow up of a PCa? Please report what is the clinical significance of BCR in introduction section
Line 48: “inert” is wrong; indolent is the correct term
Line 63: the explanation of Gleason score is too simple
Line 82: the correct number of patients is 487, not 497?
Line 82: the classification in low and high GS is an artificial decision of authors and is not corroborated by scientific data
Statistical analyses are not reported in a clear way and not easily reproducible
The characteristics of the patients included in the validation set are not reported
In discussion section there is no mention of the limitations of the study (eg small sample size)
Conclusions doesn’t report data of the study results
Reference number is too high
Author Response
Thank you for the reviewers' opinions. These comments are very helpful in improving the quality of the manuscript. We have carefully revised our manuscript to further clarify the logic of writing and improve its quality. Words in purple are the changes we have made to the manuscript. Now I'll go over the reviewers' comments point by point and highlight the changes in the revised manuscript. Full details of the files are listed. We sincerely hope that you find our responses and modifications satisfactory.
- PRAD is not a correct acronym, please modify in PCa.
Response: We have corrected all ‘PRAD’ in the manuscript to ‘PCa’.
- Line 41: “BCR is characterized by two or even more consecutively elevated serum PSA levels of 41 more than 0.2 ng/ml” it is true just for patients underwent radical prostatectomy, and not radiotherapy
Response: Thanks for the reviewers’ comments. We have revised the introduction part of the manuscript to read: "BCR is characterized by two or even more consecutively elevated serum PSA levels of more than 0.2 ng/ml in patients who underwent radical prostatectomy." (see lines 41–43)
- Why is so important the BCR during the follow up of a PCa? Please report what is the clinical significance of BCR in introduction section
Response: Thanks for the reviewers’ comments. We have reported the clinical significance of BCR in the introduction part of the manuscript as follows: " The condition of an increased serum PSA level following radical prostatectomy (RP) or radiation treatment for localized PCa is known as biochemical recurrence (BCR) or biochemical relapse[1]. BCR signifies a higher risk of metastases and PCa that is resistant to castration[2]. BCR is characterized by two or even more consecutively elevated serum PSA levels of more than 0.2 ng/ml for patients underwent radical prostatectomy[3]. Many study showed that within 5 years of receiving initial therapy, approximately 20 to 30 percent of patients with clinically localized cancer will experience a clinical recurrence[4, 5]. Clinical recurrence and metastasis are more likely to occur in BCR patients, especially early BCR patients, while the median survival for patients presenting with metastatic PCa is only 30 months[6]. Therefore, it is essential to discover more accurate biomarker for better prediction of BCR and monitoring the disease.” (see lines 38-64.)
- Line 48: “inert” is wrong; indolent is the correct term
Response: The point has been well taken. (see line 68)
- Line 63: the explanation of Gleason score is too simple
Response: We have appropriately incorporated GS explanations and mentioned references in the article as: “The GS is the sum of the grades of the first and second Gleason patterns of a primary cancer sample. It has been clearly shown that GS have been associated with BCR[7] and prostate cancer mortality [8, 9]. Considering GS is one of the best predictors of PCa prognosis, the strategy of finding differentially expressed transcripts in tumors stratified by GS is anticipated to acquire pertinent data on the potential for tumor aggressiveness[10].” (see lines 83-88).
- Line 82: the correct number of patients is 487, not 497?
Response: The correct number of patients is 487. We have made change to the article. (see line 117)
- Line 82: the classification in low and high GS is an artificial decision of authors and is not corroborated by scientific data
Response: Thanks for the reviewers’ comments. According to the current prostate cancer guidelines, the prognosis for patients with high GS (ISUP grade 4 cancers and ISUP grade 5 cancers) malignancies is still poorer than for patients with low GS (ISUP grade 3 cancers, ISUP grade 2 cancers, and ISUP grade 1 cancers). Meanwhile, numerous studies employ the same grouping method to achieve better research results[11-13].. Importantly, our findings also provide insightful data on the significance and value of prediction in PCa.
- Statistical analyses are not reported in a clear way and not easily reproducible
Response: We rewrote the part of Statistical analyses: “Student t test was used to compare the expression of genes and risk score between indicated two groups. The BCR-free survival between indicated two groups was compared by Kaplan-Meier analysis and P values were calculated with log rank test. Univariate Cox analysis and multivariate Cox regression analysis were performed to explore independent predictors of BCR-free survival. The hazard ratio (HR) and 95% confidence interval (CI) were calculated to compare the associated with BCR-free survival. ROC curves were implemented to identify the predictive accuracy of the risk signatures. All analyses were processed using the R software (v. 4.2.1) and SPSS (Version 26.0). If not specified above, P value less than 0.05 was considered statistically significant.” (see lines 157-165)
- The characteristics of the patients included in the validation set are not reported
Response: We added a description of the patient's characteristics to the training dataset and validation dataset. See Supplementary Table 1 and Supplementary Table 2.
- In discussion section there is no mention of the limitations of the study (eg small sample size)
Response: In the discussion section, we have mentioned the limitations of the study. "There are several limitations in this study that need to be acknowledged. First, as this study was conducted using retrospective data that was obtained from public datasets, and further prospective results are needed to support each other. Second, our study is the small number of cases with BCR or who died of PCa. Since BCR occurs over a wide time span, from a few months to over 15 years following the initial treatment[14], a long follow-up period is required to ascertain these outcome events. Third, future study is required to clarify the detailed molecular mechanism and function of these four genes in the development and progression of PCa. ( see lines 321-330.
- Conclusions doesn’t report data of the study results
Response: Thanks for the reviewer‘s comment, we have revised the conclusion part of the manuscript as follows: "Taken together, we conducted an integrated study to develop a GS-based four-gene signature (SFRP4, FEV, COL1A1, SULF1) for the prediction of the BCR of PCa patients after RP. The results from this study revealed a powerful prognostic indicator independent of and complementary to existing clinical factors for prognostication in PCa, such as the previously established Partin tables[15] and Kattan nomograms[16] " ( see lines 359-364)
- Reference number is too high
Response: We have appropriately deleted the references.
Supplementary table 1. Clinical characteristics for training cohorts.
|
Variables |
Training dataset (n=410) |
|
Age |
|
|
<70 |
373 |
|
>=70 |
37 |
|
pT |
|
|
<=T2 |
150 |
|
>=T3 |
255 |
|
Unknown |
5 |
|
pN |
|
|
N0 |
291 |
|
N1 |
67 |
|
Unknown |
52 |
|
GS |
|
|
>=8 |
170 |
|
<8 |
240 |
|
BCR |
|
|
Yes |
48 |
|
No |
362 |
|
Time to BCR |
0.90-148.52 |
Supplementary table 2. Clinical characteristics for validation cohorts.
|
Variables |
Validation dataset (n=92) |
|
pT |
|
|
<=T2 |
48 |
|
>=T3 |
42 |
|
Unknown |
2 |
|
Surgical margins |
|
|
Negative |
50 |
|
Positive |
42 |
|
GS |
|
|
>=8 |
15 |
|
<8 |
75 |
|
Unknown |
2 |
|
BCR |
|
|
Yes |
45 |
|
No |
47 |
|
Time to BCR |
0.36-103,43 |
- Zhang, W. and K. Zhang, A transcriptomic signature for prostate cancer relapse prediction identified from the differentially expressed genes between TP53 mutant and wild-type tumors. Sci Rep, 2022. 12(1): p. 10561.
- Antonarakis, E.S., et al., The natural history of metastatic progression in men with prostate-specific antigen recurrence after radical prostatectomy: long-term follow-up. BJU Int, 2012. 109(1): p. 32-9.
- Moul, J.W., Prostate specific antigen only progression of prostate cancer. J Urol, 2000. 163(6): p. 1632-42.
- <Management of Biochemically Recurrent Prostate Cancer After Local Therapy- Evolving Standards of Care and New Directions.pdf>.
- Briganti, A., et al., Patterns and predictors of early biochemical recurrence after radical prostatectomy and adjuvant radiation therapy in men with pT3N0 prostate cancer: implications for multimodal therapies. Int J Radiat Oncol Biol Phys, 2013. 87(5): p. 960-7.
- Guo, J., et al., A novel 8-gene panel for prediction of early biochemical recurrence in patients with prostate cancer after radical prostatectomy. Am J Cancer Res, 2022. 12(7): p. 3318-3332.
- Bibikova, M., et al., Expression signatures that correlated with Gleason score and relapse in prostate cancer. Genomics, 2007. 89(6): p. 666-72.
- Sinnott, J.A., et al., Prognostic Utility of a New mRNA Expression Signature of Gleason Score. Clin Cancer Res, 2017. 23(1): p. 81-87.
- Penney, K.L., et al., mRNA expression signature of Gleason grade predicts lethal prostate cancer. J Clin Oncol, 2011. 29(17): p. 2391-6.
- Humphrey, P.A., Gleason grading and prognostic factors in carcinoma of the prostate. Mod Pathol, 2004. 17(3): p. 292-306.
- Kurokawa, R., et al., Osteolytic or mixed bone metastasis is not uncommon in patients with high-risk prostate cancer. Eur J Radiol, 2022. 157: p. 110595.
- Wegener, D., et al., Postoperative Radiotherapy of Prostate Cancer: Adjuvant versus Early Salvage. Biomedicines, 2022. 10(9).
- Nishimoto, M., et al., Prognostic factors in Japanese men with high-Gleason metastatic castration-resistant prostate cancer. Transl Cancer Res, 2022. 11(8): p. 2681-2687.
- Popiolek, M., et al., Natural history of early, localized prostate cancer: a final report from three decades of follow-up. Eur Urol, 2013. 63(3): p. 428-35.
- Eifler, J.B., et al., An updated prostate cancer staging nomogram (Partin tables) based on cases from 2006 to 2011. BJU Int, 2013. 111(1): p. 22-9.
- Stephenson, A.J. and M.W. Kattan, Nomograms for prostate cancer. BJU Int, 2006. 98(1): p. 39-46.
Round 2
Reviewer 2 Report
Authors answered to all questions.